# Utility of Dual-Energy Computed Tomography in Clinical Conundra

**DOI:** 10.3390/diagnostics14070775

**Published:** 2024-04-07

**Authors:** Ahmad Abu-Omar, Nicolas Murray, Ismail T. Ali, Faisal Khosa, Sarah Barrett, Adnan Sheikh, Savvas Nicolaou, Stefania Tamburrini, Francesca Iacobellis, Giacomo Sica, Vincenza Granata, Luca Saba, Salvatore Masala, Mariano Scaglione

**Affiliations:** 1Department of Emergency Radiology, University of British Columbia, Vancouver General Hospital, Vancouver, BC V5Z 1M9, Canadaismail.ali@vch.ca (I.T.A.);; 2Department of Radiology, Ospedale del Mare-ASL NA1 Centro, Via Enrico Russo 11, 80147 Naples, Italy; 3Department of General and Emergency Radiology, A. Cardarelli Hospital, Via A. Cardarelli 9, 80131 Naples, Italy; iacobellisf@gmail.com; 4Department of Radiology, Monaldi Hospital, Azienda Ospedaliera dei Colli, 80131 Naples, Italy; gsica@sirm.org; 5Division of Radiology, Istituto Nazionale Tumori IRCCS Fondazione Pascale—IRCCS Di Napoli, 80131 Naples, Italy; 6Medical Oncology Department, AOU Cagliari, Policlinico Di Monserrato (CA), 09042 Monserrato, Italy; 7Department of Medicine, Surgery and Pharmacy, University of Sassari, Viale S. Pietro, 07100 Sassari, Italy; samasala@uniss.it (S.M.);; 8Department of Radiology, Pineta Grande Hospital, 81030 Castel Volturno, Italy; 9Department of Radiology, James Cook University Hospital, Marton Road, Middlesbrough TS4 3BW, UK

**Keywords:** dual energy scanned projection radiography, dual energy CT application, dual energy CT system, gastrointestinal, genitourinary, musculoskeletal, neuroradiology

## Abstract

Advancing medical technology revolutionizes our ability to diagnose various disease processes. Conventional Single-Energy Computed Tomography (SECT) has multiple inherent limitations for providing definite diagnoses in certain clinical contexts. Dual-Energy Computed Tomography (DECT) has been in use since 2006 and has constantly evolved providing various applications to assist radiologists in reaching certain diagnoses SECT is rather unable to identify. DECT may also complement the role of SECT by supporting radiologists to confidently make diagnoses in certain clinically challenging scenarios. In this review article, we briefly describe the principles of X-ray attenuation. We detail principles for DECT and describe multiple systems associated with this technology. We describe various DECT techniques and algorithms including virtual monoenergetic imaging (VMI), virtual non-contrast (VNC) imaging, Iodine quantification techniques including Iodine overlay map (IOM), and two- and three-material decomposition algorithms that can be utilized to demonstrate a multitude of pathologies. Lastly, we provide our readers commentary on examples pertaining to the practical implementation of DECT’s diverse techniques in the Gastrointestinal, Genitourinary, Biliary, Musculoskeletal, and Neuroradiology systems.

## 1. Introduction

Single-Energy CT (SECT) has long been validated as the standard imaging modality in acute clinical settings [1]. Most SECT scans are acquired at 120 kVp, producing a poly-energetic X-ray beam with an average energy of 75 keV and an energy peak of 120 kVp [2]. SECT has become a first-line tool in emergency departments due to its accessibility and high diagnostic accuracy [3]. However, in certain situations, the diagnostic limitations of SECT are evident due to multiple inherent factors including the phase of the study and the inability to separate different materials that possess similar attenuation on SECT. Dual-Energy CT (DECT) has been used in clinical practice since 2006 [4]. It has emerged as a promising tool with multiple clinical applications already demonstrated. DECT can shed light on some of the diagnostic challenges encountered when using conventional SECT, aiding radiologists and physicians to problem-solve clinical conundrums and facilitate patient management. 

## 2. Principles of Attenuation

CT images represent the attenuation of X-ray photons by different body constituents, with attenuation expressed in Hounsfield Units (HU) [5]. Attenuation refers to the weakening of X-ray photons as they pass through a material and the attenuation of a specific material is related to its density, effective atomic number (Z), and the energy level of the incoming X-ray beam [5]. The linear attenuation coefficient (µ) is a material-specific constant quantifying the fraction of attenuated incident X-ray photons per unit thickness of that material. It is directly proportional to (Z) and increases with increasing physical density of the absorbing material. This linear coefficient can be normalized as the mass attenuation coefficient per unit density of a material to produce a value that is constant for a given element independent of the density of the material.

Two dominant physics principles account for X-ray attenuation; the Photoelectric effect and Compton effect. The Photoelectric effect is one of the principal forms of interaction of X-ray photons with matter. The incident photon interacts with and ejects an inner shell electron of an atom. The probability of the effect is at its maximum when the energy of the incident photon is equal to or just greater than the binding energy of the electron in its shell (known as the k-edge value). The removed electron is called a photoelectron and the incident photon is completely absorbed. The photoelectric effect is directly proportional to (Z) of a material and inversely proportional to the incident photon’s energy, i.e., the probability of the photoelectric effect increases with higher (Z) and lower energy levels.

The most abundant elements in human tissue have k-edges too low to be detectable ranging from 0.1 keV to 0.53 keV [6]. Although calcium has a relatively low k-edge value (4 keV), its high value relative to soft tissue causes markedly greater attenuation of low-energy photons producing inherent image contrast. Iodine has a k-edge value of 33.2 keV, within the range of energies used in diagnostic imaging, making it ideal for use in contrast media due to a combination of mass attenuation coefficient and the photoelectric effect. 

SECT is limited by its sole reliance on CT HU for quantitation [7]. This explains why, depending on their relative densities, two materials of differing composition such as calcium and iodine may yield the same measurable HU when subjected to a single X-ray beam but produce different HU values when exposed to different energy levels at DECT, independent of their density value. Because iodine and calcium have high (Z), they are more susceptible to the photoelectric effect. Contrarily, soft tissue demonstrates a weak photoelectric effect and less variation of attenuation values with different energy levels [8]. 

The Compton effect is the main cause of scattered radiation. It occurs due to the interaction of the incident photon with free unattached electrons or loosely bound outer shell electrons. Unlike the photoelectric effect, the Compton effect is relatively independent of the photon’s energy level.

## 3. Principles of DECT 

DECT assesses attenuation by materials when submitted to high (140–150 kVp) and low energy (80–100 kVp) photon beams and acquires two datasets for the imaged anatomical range. Post-processing of dual-energy data can be conducted either before (in the projection-space domain) or after (in the image-space domain) the reconstruction of high- or low-energy images, depending on the scanner [9,10]. Postprocessing manipulation is utilized to reconstruct three different types of images: mixed images, material-specific images, and virtual monoenergetic images [11]. 

Images derived from a combination of the high and low-energy datasets are created and are called mixed images. Using varying percentages of the high and low kVp datasets, the resultant images simulate SECT acquired at 120 kVp [12] and are sent to the Picture Archiving and Communication System (PACS) for interpretation. 

DECT material-specific images are created after evaluating the interaction of all body constituents with high and low energy levels. These images are generated through differential attenuation analysis to obtain a dual-energy index. Postprocessing software is used to calculate the attenuation properties of each voxel at low and high energy and a mathematical algorithm is used to determine the proportion of dominant materials within the voxel based on a three-material decomposition technique to evaluate the representation of iodine, calcium, or fat in each voxel [13]. Once the proportions are known, different sets of images are generated after subtracting each constituent. 

Virtual non-contrast (VNC) images, sometimes referred to as virtual unenhanced (VUE) images depending on the vendor, are constructed after the voxels containing iodine are excluded from the image. VNC images help in avoiding the need for extra time and radiation dose associated with acquiring a separate true non-contrast scan. These advantages are particularly evident in multiphasic studies such as CT angiography, CT urography, or multiphasic liver CT, where the necessity for a true non-contrast scan can be omitted. This can result in reductions in radiation dose ranging from 19% to 60%, depending on the specific protocol used [14,15,16,17,18,19].

Likewise, iodine-only images and virtual non-calcium (VNCa) images can be generated. It is also possible to superimpose a color-coded iodine overlay map (IOM) to quantify the amount of iodine in a specific region of interest.

DECT allows the extrapolation of images for various single-energy levels. The resultant virtual monoenergetic image (VMI) simulates the image that would be obtained if the scanner were to emit a true monoenergetic beam of X-rays. Using low-energy images improves the contrast-to-noise ratio by increasing the attenuation of materials with high atomic numbers such as iodine at the expense of decreased spatial resolution and increased noise. This facilitates the detection of iodine-containing structures such as contrast-opacified vessels and hypervascular tumors rendering them more conspicuous [20]. It also enables a decrease in the volume of intravenous iodinated contrast administered which can be useful in patients with reduced renal function [21]. For instance, without compromising diagnostic accuracy, reductions in iodinated contrast dosage of up to 50% can be attained by utilizing 40- and 50-keV monoenergetic images in abdominal aorta CT angiography [22,23,24,25]. Similarly, reductions of 50% in iodinated contrast dosage for CT urography and 40% for coronary CT angiography are also feasible [26,27]. 

Conversely, high-energy images improve signal-to-noise ratio and ameliorate the effect of streak artifact but with decreased attenuation of body constituents at higher energies, thereby limiting soft tissue contrast resolution.

DECT postprocessing application also utilizes Rho-Z Maps (Z effective) to reconstruct images based on the material decomposition of substances. The Rho Z decomposition technique involves separation of materials based on their effective number (Rho) and electron density (z) [28]. Through material composition analysis, this application can identify and characterize different materials depending on their relative atomic number. This can be useful in assessing enhancement of an imaged radiodense structure or confirming absence of iodine within the structure [20].

## 4. DECT Systems

### 4.1. Dual-Source

Dual-source scanner with dual detector arrays utilizes two X-ray sources and two data-acquisition systems mounted on the gantry in an orthogonal fashion [29]. Each X-ray source is equipped with its own high-voltage generator allowing independent control of the X-ray tube potential and current. High-energy dataset is obtained at 120 or 140 kVp and low-energy dataset acquired at 80 or 100 kVp [30]. Using two separate X-ray sources allows beam filtration and current modulation in each tube resulting in optimization of image quality [30]. However, the dual-source images suffer from limited temporal and spatial registration because the high and low-energy scans are acquired at slightly different times. Furthermore, the available field of view is limited to 33cm, somewhat hindering evaluation in patients with large body habitus [30]. 

### 4.2. Single-Source with Fast Kilovoltage Switching

This prototype employs a single source scanner with a single detector layer. It is contingent on a single X-ray source with fast switching between two kilovoltage settings (80 and 140 kVp) at intervals of 0.5 ms during a single gantry rotation to generate high and low-energy X-ray acquisitions [30]. Unlike the dual-source scanner, the tube current remains constant for both energy levels and cannot be altered simultaneously. The exposure time ratio is varied between the 80 and 140 kVp acquisitions (typically 65% of exposure time at 80 kVp and 35% at 140 kVp) in order to optimize the contrast-to-noise ratio. The detector system has a fast-sampling capability to acquire the alternating high and low-energy data. The advantages of the fast kilovoltage switching scanner include good temporal and spatial registration [30]. Furthermore, the available field of view is increased to 50 cm for image analysis. Drawbacks include limited spectral separation between the high and low-energy datasets [30]. Additionally, the noise-to-signal ratio is higher on images obtained at lower energy levels as the tube current cannot be modulated at the same time the peak voltage is altered [30].

### 4.3. Single-Source with Dual-Layer Detector

This scanner detector system is modified with two scintillation layers arrayed one on top of each other to receive separate high and low-energy data from a single X-ray source [30]. The top detector layer captures low-energy data whilst the bottom layer captures high-energy data. This prototype offers excellent temporal and spatial registration but has limited energy separation with substantial spectral overlap.

### 4.4. Single-Source with Twin-Beam Filtration

This system utilizes a single energy source with a split filter placed in the collimator. The filter is composed of gold and tin arranged in close proximity along the longitudinal axis to facilitate separation of the high and low-energy X-ray spectra. The respective halves of the detector capture these distinct energy spectra [31,32]. By employing tin and gold, the filter segregates the low and high-energy components of the polychromatic X-ray beam, consequently achieving spectral separation [33,34]. This system is cost-effective and does not require significant hardware modification. Installing a split filter into the tube collimator of an established CT scanner is undertaken to enable dual-energy imaging [31,32,33]. This enhancement allows for image acquisition across the entire field of view, while automatic tube current modulation ensures optimal radiation dosage for the patient [31,33].

Nonetheless, the spectral separation achieved through a split filter is relatively restricted when contrasted with the capability of a dual-source dual-energy scanner [32,34]. Moreover, there exists the possibility of cross-scattering, wherein one side of the beam may influence the adjacent side of the detector. Additionally, a higher power output from the X-ray tube becomes imperative due to the beam prefiltration, posing a limitation particularly in imaging obese patients. Furthermore, temporal discrepancies between low- and high-energy data are observed [31].

### 4.5. Single-Source with Sequential (Dual-Spin) Acquisition

Sequential acquisition, proposed as one of the initial methodologies for obtaining dual-energy images without specialized hardware, involves scanning the entire volume sequentially at low and high kilovoltage settings [34]. The system functions in either an axial or helical mode. In the axial mode, two gantry rotations, each at a different energy level, are acquired at each table position before the table moves in the z-axis dimension [7]. In the helical mode, the entire scan volume undergoes scanning with a helical acquisition at one energy level, immediately followed by a helical acquisition of the same scan volume at another energy level. As two acquisitions are performed, each acquisition is executed at half the dose typically used in a conventional 120 kVp scan to mitigate the risk of heightened radiation exposure [8].

This approach, while effective, introduces a considerable delay between acquiring the two datasets. However, this delay can be mitigated to some degree by alternating the tube potential between each successive gantry rotation [7,31,32].

No significant hardware alterations are required with this system. Any CT scanner can perform sequential scanning at two different voltages, allowing the datasets to be combined for spectral analysis [31].

By imaging at two different potentials at the same location during each gantry rotation, the views align precisely, facilitating projection space alignment and material decomposition [33]. Moreover, automatic exposure control can be applied, allowing for tube current modulation at different potentials to achieve optimal noise levels [33].

Drawbacks include spectral distortion due to patient movement between acquisitions. Additionally, implementing this method in cardiac studies and angiographic acquisitions can be challenging due to significant temporal skew, necessitating high temporal resolution [31]. Consequently, its application is primarily limited to relatively static organs and non-contrast studies [34,35]. Table 1 summarizes the strengths and weaknesses of various DECT technologies

## 5. DECT Applications

### 5.1. Examples in the Gastrointestinal System

#### 5.1.1. Bowel

Imaging of the unprepared bowel by conventional SECT may have limitations which can be resolved by using DECT. The principle of DECT for bowel pathology relies primarily on identifying whether mural enhancement is present or absent after administration of intravenous contrast [36]. It is important to avoid administering enteric iodinated contrast when DECT postprocessing techniques are used as enteric contrast would compromise assessment of mural enhancement [11].

The subtle differences in mural attenuation in inflammatory conditions of the bowel can be accentuated by employing iodine quantification postprocessing techniques. Lee et al. revealed that mucosal hyperemia related to active Crohn’s disease was most conspicuous on VMI at 40 keV, as compared to conventional SECT obtained at 120 kVp, increasing the sensitivity and negative predictive value for diagnosing active Crohn’s disease [37].

A suspected large bowel neoplasm can appear more conspicuous on IOM due to iodine uptake which can increase the confidence in raising the probability of an underlying malignant process by the reporting radiologist [11].

Bowel ischemia is ascribed to arterial thromboembolic phenomena or venous occlusion as well as hypoperfusion secondary to low-flow states. Segmental hypoenhanced or unenhanced bowel wall strongly suggests ischemia. These changes can be subtle on conventional CT. The application of low-keV VMI increases the difference in attenuation between ischemic and non-ischemic segments of bowel [11]. Similarly, IOM increases the conspicuity of non-enhancing bowel helping radiologists to confidently attribute the finding to ischemia (Figure 1). In the latter stages of ischemia, submucosal hemorrhage may ensue resulting in mural hyperattenuation, which can be mistaken for mural enhancement on conventional SECT. VNC images and IOM can be utilized to demonstrate intramural hemorrhage and decreased iodine mural uptake respectively supporting the diagnosis of bowel ischemia [11].

Gastrointestinal bleeding is caused by many conditions including bleeding disorders, anticoagulant therapy, and neoplasms [38]. VNC images can confirm the hyperattenuating nature of intramural or intraluminal blood whilst IOM helps to exclude the presence of an underlying enhancing lesion. Furthermore, IOM can increase the conspicuity of intraluminal extravasation of contrast in keeping with active bleeding and enable the differentiation of these areas of active extravasation from hyperattenuating digestive material or intestinal foreign bodies. Equally, low-keV VMI can identify active hemorrhage by promoting the conspicuity of iodine. Another advantage of contrast-enhanced DECT in the setting of gastrointestinal bleeding and bowel ischemia is its ability to create VMI derived from contrast-enhanced phases, allowing up to 30% reduction in radiation dose when compared to the triphasic studies usually performed as standard in suspected gastrointestinal bleeding or bowel ischemia [39].

#### 5.1.2. Appendix

In acute appendicitis, IOM can help increase the confidence of diagnosis by detecting areas of mural hyperenhancement owing to hyperemia; a feature that can be subtle on conventional CT in the early stages of inflammation [11]. Likewise, IOM can be utilized to depict complications of appendicitis including ischemia, gangrene, or perforation demonstrated by discontinuity of enhancement of the bowel wall. This finding can be subtle or even undetectable on conventional SECT.

### 5.2. Examples in the Genitourinary System

#### 5.2.1. Urolithiasis

Patients presenting to the emergency department complaining of acute flank pain are frequently investigated with a CT of the kidney, ureters, and bladder (CT KUB) to exclude urinary tract calculi, considering its high sensitivity (97%) and specificity (96%) [40,41]. The commonest composition of urinary tract calculi is calcium oxalate with uric acid stones constituting up to 10% of renal calculi [42]. It is of the essence to differentiate uric acid from non-uric acid stones as the former can be treated medically with urinary alkalinization [42]. Non-uric acid stones, nevertheless, are treated with fragmentation or extraction. Calcium oxalate monohydrate, cystine, and brushite calculi may require percutaneous lithotomy, considering their increased resistance to fragmentation [43].

SECT is restricted in establishing the chemical composition of urinary tract calculi, depending solely on attenuation of calculi with considerable overlap among different types of calculi [44]. DECT differentiates uric acid from non-uric acid stones with an accuracy of 90–100% [45] (Figure 2). Calculus characterization is achieved by calculating a dual-energy index after analyzing the differential attenuation of the calculus at low and high-energy levels. Second and third-generation DECT scanners employ additional filtration of high-energy photons allowing for a better spectral separation and more detailed analysis of stone composition [46]. This allows differentiation of cystine, struvite, calcium oxalate dihydrate or monohydrate, and apatite stones [47]. Moreover, VNC images can be derived from a contrast-enhanced DECT scan to identify urinary stones masked by contrast opacification [48]. It is worth noting that DECT is limited in the assessment of small stones (<3 mm) in patients with a high body mass index because of the reduced signal-to-noise ratio [49].

#### 5.2.2. Inflammation, Neoplasm and Hemorrhage

IOM increases the conspicuity of the hyperemic urothelium by accentuating iodine in cases of urinary tract infections. IOM can also be utilized to identify focal areas of decreased iodine uptake in an inflamed organ such as the prostate raising concern for a phlegmon or abscess formation [11]. VNC images can identify hyperattenuating blood products in keeping with hematoma within the collecting system and IOM can be used to identify areas of active extravasation in keeping with acute hemorrhage.

DECT analysis of renal masses is sensitive, specific and accurate at differentiating enhancing masses when using an iodine-specific postprocessing application [50]. This can be helpful in differentiating hemorrhagic or proteinaceous renal cysts from neoplasm (Figure 3).

### 5.3. Examples in the Biliary System

#### 5.3.1. Cholelithiasis

Ultrasound is distinguished for its sensitivity in the evaluation of cholelithiasis and acute cholecystitis. Despite that, CT is frequently used for the evaluation of right upper quadrant pain especially in the acute setting. Up to 57% of gallstones isoattenuate to the surrounding bile and are therefore inconspicuous on conventional SECT [51].

Unlike water, which demonstrates minimal change in attenuation when imaged by various X-ray spectra, heavy atoms such as cholesterol-containing gallstones possess a CT value that increases with increasing X-ray tube voltage [52]. Therefore, VMI can be used to identify noncalcified gallstones that appear hypoattenuating to surrounding bile at low-keV (40 keV) and hyperattenuating relative to bile at high-keV reconstruction (≥140 keV) [53] (Figure 4).

As described earlier, DECT’s material decomposition techniques allow the differentiation of materials of various chemical compositions that have similar attenuation values on SECT. These techniques with a calcium and lipid base, or a calcium, lipid, and water (three-material decomposition) base algorithm can be utilized to identify non-calcified gallstones with excellent sensitivity and specificity ranging from 80–95% and 95–100% respectively [54].

#### 5.3.2. Cholecystitis

Gallbladder hyperemia concordant with acute cholecystitis is easily discernable using low-keV VMI and IOM reconstruction images. These techniques also improve detection of gallbladder wall gangrene by depiction of areas of decreased enhancement in cases of complicated cholecystitis [55].

### 5.4. Examples in the Musculoskeletal System

#### 5.4.1. Bone Marrow Edema (BME)

Accurate and early identification of acute fractures is important to avoid complications including malunion, osteonecrosis, degenerative arthritis, persistent pain, and functional compromise [56]. BME typically occurs in response to a fracture and is attributable to the accumulation of fluid or blood within the bone marrow. In acute settings, BME can be key in the identification of non-displaced fractures which may be subtle or even impossible to identify on SECT.

Magnetic Resonance Imaging (MRI) is the preferred method for identifying BME [57]. However, due to potential contraindications and limited availability of MRI especially in the acute setting, CT remains the most frequent modality used in trauma.

DECT enables BME detection using the three-material decomposition algorithm, determining the contributions of fat, soft tissue, and calcium within each voxel. The calcium is removed from the data to produce a virtual non-calcium image (VNCa) representative of underlying bone marrow. Increased attenuation of the bone marrow can then be qualitatively assessed using color-coded VNCa maps [58] (Figure 5 and Figure 6). Comparing color signals with the adjacent region of the same bone or with the contralateral side is important for the detection of an abrupt change in attenuation raising concern for BME.

#### 5.4.2. Gout

Gout is the commonest and the most validated DECT application in the musculoskeletal system with multiple studies showing sensitivities ranging between 78–100% and specificities of 89–100% [59,60,61]. In fact, a positive DECT scan is part of the 2015 American College of Radiology classification criteria for gout [62]. DECT utilizes two-material (uric acid and calcium) or three-material (uric acid, calcium and soft tissue) decomposition techniques for the characterization of monosodium urate (MSU) crystals. The separation of materials is feasible because the attenuation of the low atomic weight MSU crystals is different from that of the high atomic weight calcium when exposed to different energy levels [63]. The MSU crystals are color-coded and overlaid on gray-scale images as illustrated in Figure 7. The quantification of the volume of crystal burden can be accurately assessed to estimate the severity of the disease, prognosis, and monitoring response to target therapy [59,64].

### 5.5. Examples in Neuroradiology

#### Post Endovascular Thrombectomy (EVT)

Acute ischemic stroke (AIS) is the leading cause of disability and mortality worldwide [65]. EVT is recommended as first-line treatment for AIS with large artery occlusion [66,67,68]. Blood–brain barrier injury secondary to ischemia commonly causes post-interventional cerebral hyperdensities (PICHs) on non-contrast CT Brain studies performed after EVT [68]. PICHs are usually ascribed to contrast staining (CS) secondary to limited vascular endothelial injury. Nevertheless, a serious complication of AIS is hemorrhagic transformation (HT) which is associated with worsening prognosis or even death. Hence, it is crucial to identify the cause of PICHs after EVT.

DECT is invaluable in differentiating the cause of PICHs as either due to CS or dangerous HT (Figure 8). Differentiating contrast from hemorrhage is also vital to gear the physician’s decision-making as to whether and/or when to commence anti-platelet therapy post-EVT. Utilization of VNC and IOM images can differentiate the source of PICHs as demonstrated in Figure 8.

### 5.6. Pulmonary Examples

Utilizing low keV VMI achieves a significant increase in iodine attenuation thereby enhancing the visibility of iodine-containing structures and helps distinguish enhancing from non-enhancing lesions [69]. Employing VNC imaging may result in a notable decrease in the overall administered contrast dose, as pre-contrast scans become unnecessary [69]. Additionally, high-energy VMI can be utilized to mitigate beam hardening artifacts typically observed in the thorax due to dorsal spine stabilization for scoliosis [70,71,72].

DECT is effective in identifying acute pulmonary embolism (PE). Its primary advantage lies in its capability to detect perfusion abnormalities in the lung parenchyma resulting from pulmonary arterial occlusion and to directly visualize arterial filling defects [73,74,75,76,77]. The color-coded IOM is useful to identify dark wedge-shaped segments consistent with PE. DECT is particularly beneficial in identifying small vessel occlusive emboli. One study emphasized the effectiveness of VMI in cases of suboptimal scans due to technical issues or mistimed contrast bolus injections [78].

DECT can aid in distinguishing between benign and malignant pulmonary nodules and masses [79]. It has been shown that subjective image quality for visualizing lung carcinoma is enhanced when using energy levels between 55–70 keV as opposed to conventional CT scans [80,81]. Additionally, DECT can be employed for cases involving anterior mediastinal lesions to differentiate thymic cysts from thymic epithelial tumors [79].

### 5.7. Cardiac Evaluation

The precise and swift diagnosis of patients presenting with chest pain is imperative for initiation of appropriate acute management and minimizing risk of complications [82].

Innovations in cardiac CT imaging have facilitated prompt and precise, non-invasive, evaluation of various cardiac conditions. Despite its notable negative predictive power, obtaining high resolution cardiac images presents inherent challenges, most notably cardiac motion artefact secondary to an elevated heart rate [82,83].

Various applications of DECT in cardiac imaging have demonstrated clinical feasibility and usefulness through utilization of Iodine myocardial perfusion maps, VMI and VNC reconstructions [84,85,86]. These techniques have been proven to enhance image quality and diagnostic accuracy.

Of note, only dual-layer and dual-source DECT technologies enable near complete simultaneous data acquisition, making them particularly valuable for cardiac imaging [87]. Other techniques are at risk for misregistration posing diagnostic challenge for evaluation of atherosclerotic plaque in tiny and rapidly moving coronary arteries.

Perfusion maps highlight iodine distribution within the myocardium. This allows qualitative detection and delineation of myocardial perfusion defects [83]. Studies have shown that perfusion maps increase the diagnostic accuracy for infarct detection compared to coronary computed tomography angiography (CCTA) [88,89]. Furthermore, there is evidence that perfusion maps help radiologists predict if a coronary artery stenosis is hemodynamically significant, since it can give information regarding a potential decrease in iodine uptake in the respective supplied territory [88,90,91]. Other authors demonstrated additional value of perfusion maps for detection of late-enhancing tissue, indicating chronic ischemia or scarred myocardium and potentially allowing differentiation between chronic or reversible myocardial ischemia [92,93,94].

Low KeV VMI (less than 80 KeV) improves conspicuity of iodine in comparison to conventional CT with objective increase in image quality [83]. Studies have reported arterial attenuation values exceeding 1000 HU in CT angiography when employing ultra-low keV VMI, leading to increased contrast-to-noise and signal-to-noise ratios [95,96].

Reconstructions using low-energy VMI are particularly beneficial in scenarios where contrast bolus is suboptimal. This may occur due to factors such as inaccurate ROI placement, myocardial insufficiency, or difficulties in venous access during the injection process [83].

Cardiac imagers can deliberately leverage VMI enhancing attenuation capability to decrease the necessary dosage of contrast media administered [97].

VMI has the potential to alleviate various artifacts frequently encountered in cardiac CT e.g., beam-hardening artifacts, which mimic myocardial hypoattenuation seen in ischemia, can be mitigated quantitively and qualitatively [98]. Additionally, calcium blooming artifacts, commonly observed in patients with severe calcifications, often diminish the diagnostic accuracy of CCTA exams, frequently resulting in overestimation of coronary artery stenosis [83]. There is supporting evidence indicating that high-energy VMI algorithms (such as those at 110 keV and higher) efficiently diminish blooming artifacts [99,100].

Virtual calcium subtraction is another technique which has been demonstrated to enhance the reader’s diagnostic certainty when evaluating coronary artery stenosis in individuals with extensively calcified coronary arteries [101].

### 5.8. Vascular Examples

An especially enticing application of DECT is direct CT angiography. In this method, the dual-energy algorithm detects and eliminates bone, enabling direct visualization of iodinated vessels [102,103,104,105]. Atherosclerotic calcified plaque can obscure the vessel lumen. Calcium subtraction application can assess vascular lumen patency and delineates areas of stenosis on maximum intensity projections of CT angiograms [106,107,108].

The typical CT arteriography protocol used to evaluate acute aortic syndromes involves acquiring unenhanced images to identify intramural hematomas or intimal calcifications which lead to significant radiation exposure. Despite the fact that VNC images exhibit slightly more noise compared to conventional unenhanced images, they provide diagnostic value in nearly 95% of cases [109].

DECT is also beneficial in aortic evaluation after endovascular stent repair. IOM can be utilized to distinguish high attenuation material caused by blood, bone, or contrast agent and increase confidence for endoleak detection particularly when subtle [110].

VMI at 40 keV offers enhanced contrast attenuation of poorly opacified liver veins in comparison to blended images [111]. Additionally, low keV VMI has demonstrated improvements in assessing portal vein thrombosis [112]. Both IOM and VMI at 40 keV significantly elevate diagnostic confidence and accuracy in identifying and distinguishing venous thrombosis from iodine reflux when compared to mixed (blended) images [113].

### 5.9. Oncology Examples

DECT proves valuable in distinguishing hemorrhagic brain metastasis from intracranial hemorrhage [114]. Employing a monoenergetic beam of 40 keV enhances tumor delineation. DECT can also discern hyperdense bleeding from enhancement within a hemorrhagic mass. It has been proposed that 65 keV VMI offer optimal overall image quality, while 40 keV VMI images offer superior tumor delineation [115].

Several studies have indicated that hyperattenuating and hypervascular liver lesions are more conspicuous at lower keV levels compared to conventional CT scanners [116,117].

In individuals with pancreatic adenocarcinoma, 55-keV VMI has been shown to enhance the visualization of tumors and vascular infiltration compared to blended reconstructions [118]. However, 70 keV was identified as the most favorable subjective energy level overall [119].

For delineating renal cell carcinoma tumors, the optimal keV is 40 [120]. Other studies have aimed to establish optimal thresholds for distinguishing between vascular and nonvascular renal lesions, recommending energy levels of 40–60 keV [121].

An iodine concentration of 2.0 mg/mL has been identified as the optimal threshold for discriminating between lymphoma and lymph node metastasis, with a sensitivity of 87% and specificity of 89% [122].

## 6. Challenges and Pitfalls

DECT has limitations despite its advantages. The technology is expensive and requires specialized hardware and software. Implementation of DECT requires training to staff radiologists and technologists. Additionally, the generated image datasets can be overwhelming and may turn into an incumbrance for the reporting radiologist. This challenge can be overcome by limiting data sent to PACS to the essential required for reporting.

There is paucity of literature on the cost effectiveness of DECT [123,124]. Furthermore, challenges exist for both qualitative and quantitative interpretation of images produced by DECT [125,126]. For instance, calcium in tissues may appear smaller on VNC images compared to true non-contrast imaging; potentially leading to the oversight of small calcifications or calculi [127]. In patients with chronic parenchymal liver disease, the presence of liver fat can complicate the quantification of liver iron deposition [127].

Another significant limitation is the variability in technology, terminology, and image datasets among different vendors, which hampers the standardization of quantitative metrics and poses difficulties for multi-center research studies.

Finally, the emergence of photon-counting CT holds significant promise and presents numerous benefits compared to DECT. These advantages encompass enhanced spectral resolution, refined tissue characterization, minimized image artifacts, and improved image quality [128].

## 7. Conclusions

Advancing medical technology continuously revolutionizes and reforms our ability to diagnose multiple disease processes. Various DECT applications have stood firm as problem-solving tools supporting radiologists to confidently diagnose a myriad of pathologies and help physicians to select appropriate management plans to ensure delivery of best quality care for their patients.

## Figures and Tables

**Figure 1 diagnostics-14-00775-f001:**
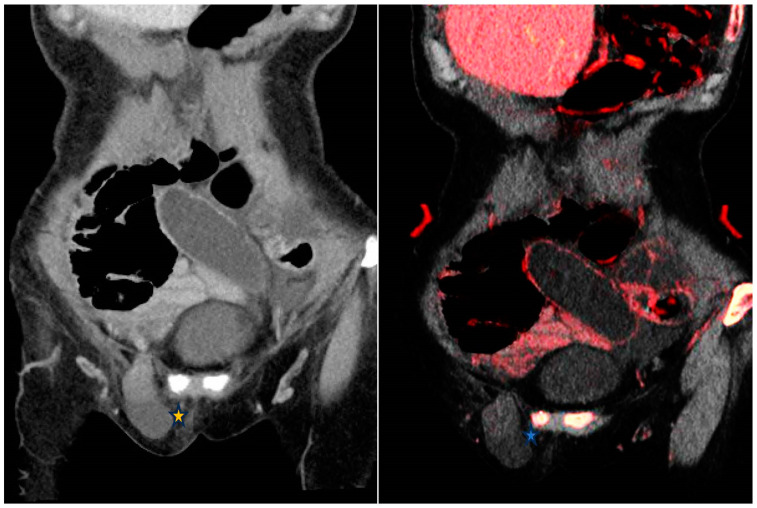
From left to right—Contrast enhanced conventional 120 kVp and iodine overlay map (IOM) coronal images of the abdomen. Closed-loop small bowel obstruction secondary to a right inguinal hernia (*****). The small bowel loop demonstrates no mural enhancement; more conspicuous on the IOM (*****) which is highly suggestive of an incarcerated hernia.

**Figure 2 diagnostics-14-00775-f002:**
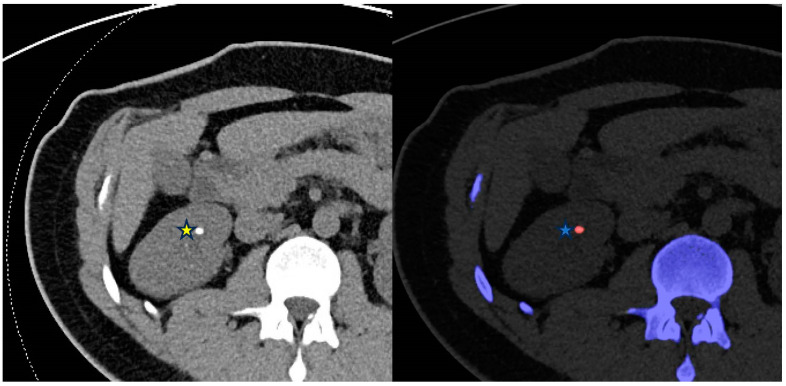
From left to right—Conventional CT KUB and DECT renal stone analysis axial images of the abdomen. The right midpole calculus (*****) is identified on the color-coded stone analysis DECT application as a uric acid calculus (*****) which can be treated medically with urine alkalinization.

**Figure 3 diagnostics-14-00775-f003:**
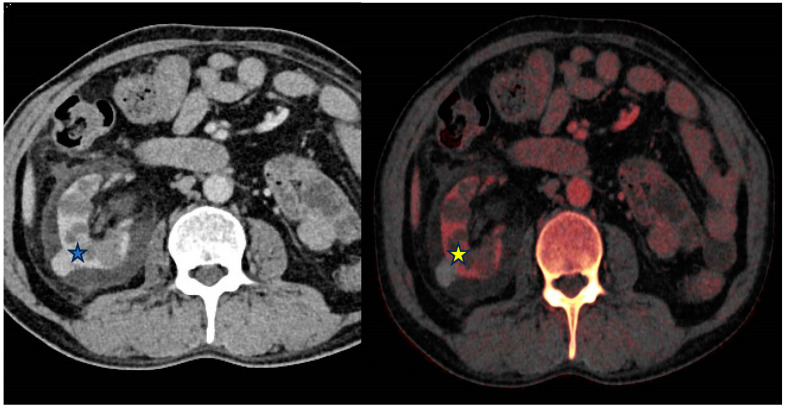
From left to right—Contrast-enhanced (portal venous) and Iodine overlay map (IOM) axial images of the abdomen. Right exophytic hyperattenuating lesion (*****) is further characterized as lacking internal enhancement on the IOM (*****) image and is therefore likely to represent a benign hyperdense cyst rather than an enhancing lesion.

**Figure 4 diagnostics-14-00775-f004:**
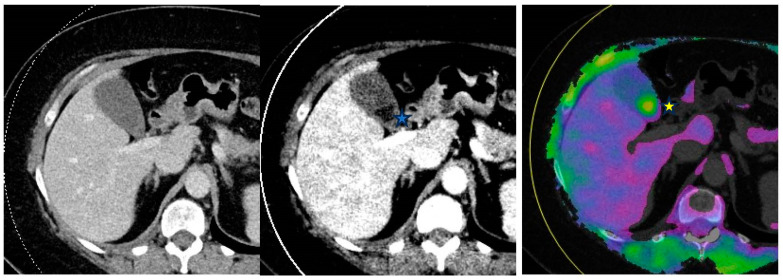
From left to right—Conventional mixed 120 kVp equivalent, 40 keV virtual monoenergetic and gallstone analysis application axial images through the abdomen. Gallstones are not visible on the conventional image as they isoattenuate to the surrounding bile. However, they are visible on the DECT images and appear hypodense (filling defect) on the virtual monoenergetic image (*****) and hyperdense (*****) on the gallstone analysis application.

**Figure 5 diagnostics-14-00775-f005:**
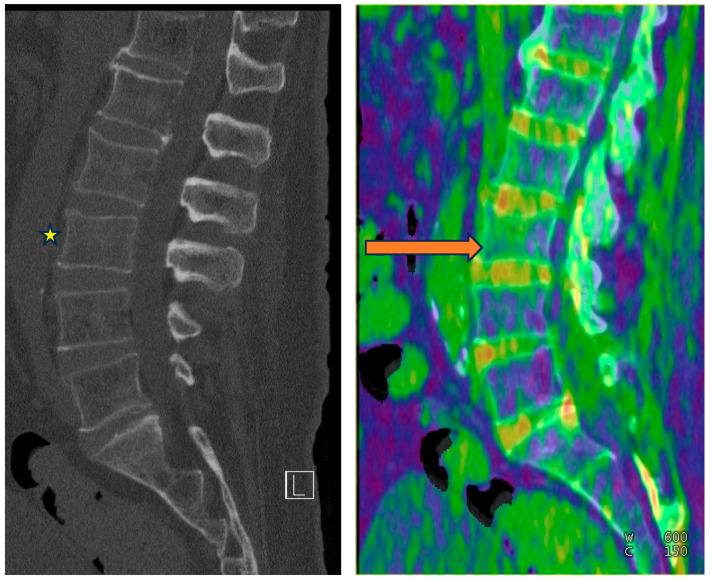
From left to right—Conventional mixed 120 kVp equivalent and bone marrow edema (BME) overlay sagittal images of the lumbar spine. Subtle L3 superior endplate compression fracture is difficult to detect on the conventional images(*****). BME is, however, confidently demonstrated on the overlay map (orange arrow).

**Figure 6 diagnostics-14-00775-f006:**
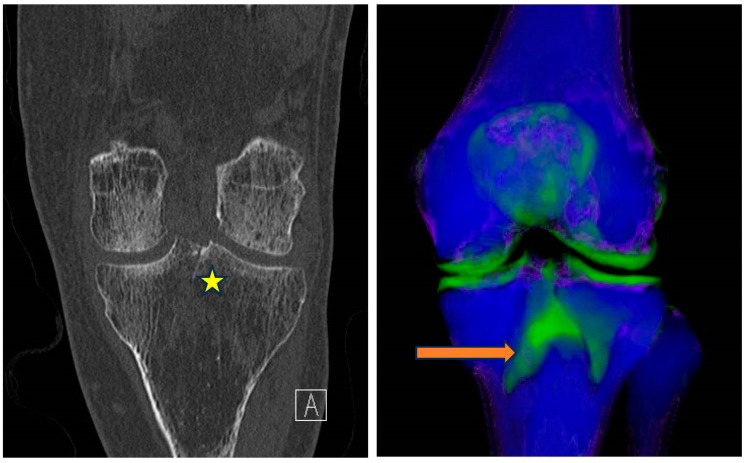
From left to right—Conventional mixed 120 kVp equivalent and 3D bone marrow edema overlay coronal images of the right knee. Subtle intercondylar tibial fracture extending into the tibial condyle (*****) is confidently diagnosed when appreciating the associated BME on the overlay image (orange arrow).

**Figure 7 diagnostics-14-00775-f007:**
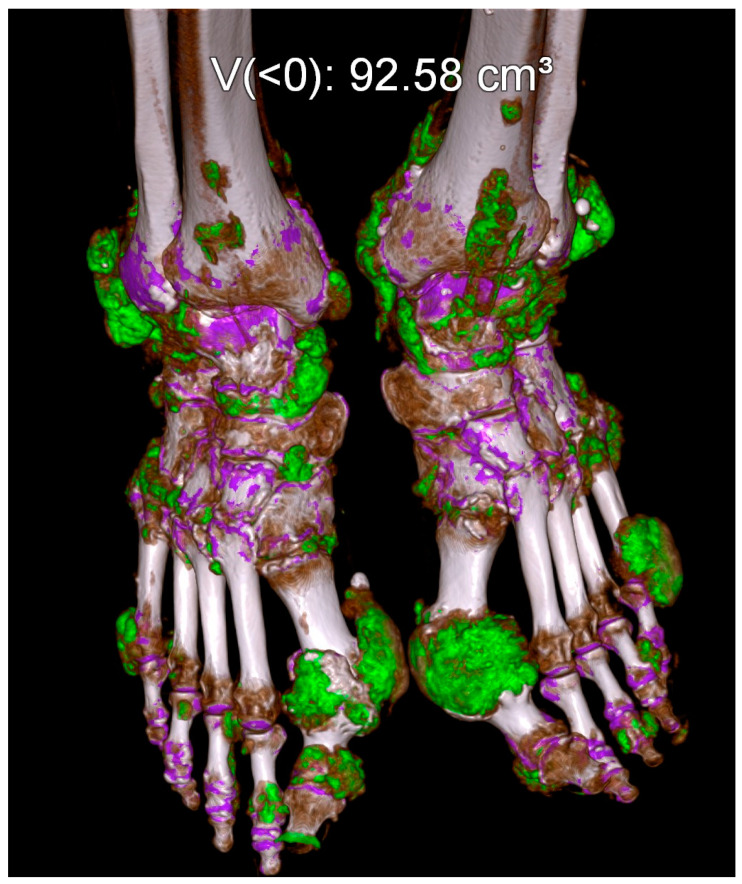
DECT gout application readily identifies the color-coded monosodium urate (MSU) crystals (green color) and assesses disease burden.

**Figure 8 diagnostics-14-00775-f008:**
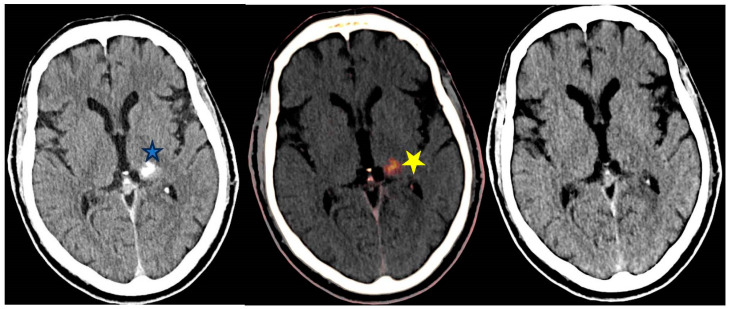
From left to right—Unenhanced conventional 120 kVp equivalent, iodine overlay map (IOM) and virtual non-contrast (VNC) axial images of the brain post Endovascular thrombectomy (EVT). Hyperdensity within the left basal ganglia on the conventional image (*****) could be caused by contrast staining (CS) or hemorrhagic transformation (HT). The hyperdensity persists on the IOM (*****) but is not detectable on the VNC image; confirming this to be secondary to CS and not HT.

**Table 1 diagnostics-14-00775-t001:** Strengths of Weaknesses of various DECT systems.

Technology	Advantages	Disadvantages
Dual source	Optimized image quality	Limited temporal and spatial resolution Small FOV *
Rapid kVp switching	Good temporal and spatial registration	Limited spectral separation High noise-to-signal ratio of low-energy images Requires specialized hardware
Dual layer detectors	Excellent temporal and spatial resolution	High spectral overlap Requires specialized hardware
Twin-beam filtration	Cost-effective Full spectral FOV * available for image acquisition	Temporal discrepancy between high- and low-energy data Cross-scatter Intrinsic lower energies due to filtration Overlapping spectra in center and edge of the beam
Dual spin	Cost-effective (can be performed on any scanner) Optimized tube current modulation Full spectral FOV * No cross scatter	Spectral distortion secondary to motion artefact

* FOV—Field of view.

## Data Availability

No new data were created or analyzed in this study. Data sharing is not applicable to this article.

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
