# Peer review of "Utility of Dual-Energy Computed Tomography in Clinical Conundra"

_diagnostics, 2024, doi:10.3390/diagnostics14070775_

Round 1

Reviewer 1 Report

Comments and Suggestions for Authors

Make the text more vendor neutral. It now reads as dedicated to Siemens scanners and Siemens postprocessing techniques. Z-effective reconstructions are lacking. No literature? No experience? No value?

Paragraph 4: add a table with strengths and weaknesses of different systems / approaches

Paragraph 4: consider adding twin beam, consider adding dual rotation

Paragraph 4: Add references to the described strengths and weaknesses

Paragraph 5: I seems that DECT and especially the evidence in the literature is highly disappointing. We are 19 years after the introduction of these systems and there is almost no evidence on the added confidence, diagnostic value and patient impact. This probably shows the limitations of the method, that we are provided with a pictorial essay.

Paragraph 5: Several statements have no reference. Please add, if possible, or state “based on our experience”.

Paragraph 5: It is a fly-through, certainly not a comprehensive overview of clinical problems. What is lacking are the vascular and pulmonary applications, all kinds of benign and malignant diseases in the described regions, iodine contrast boosting and contrast reduction. It is fair to acknowledge this somewhere and lead the readers to other reviews and essays on this topic.

Paragraph 5: It feels strange to publish another pictorial essay on dual energy CT, while photon counting CT is already on the market. I would propose to end with a paragraph related to challenges of DECT that maybe solved with photon counting detectors.

Author Response

  • Make the text more vendor neutral. It now reads as dedicated to Siemens scanners and Siemens postprocessing techniques. Z-effective reconstructions are lacking. No literature? No experience? No value?

Thank you for your comment and for bringing this to my attention. Other DECT technologies are now added to section 4 including single-source with twin beam filtration and single source with dual-spin (sequential) acquisition. Post-processing techniques in Section 3 have been updated to cover projection-space and image-space domains. Furthermore, description for Virtual Unenehanced (VUE) imaging and Rho-Z maps (Z-effective reconstruction) are now stipulated.

  •  Paragraph 4: add a table with strengths and weaknesses of different systems / approaches

Thank you for the comment. Section 4 now encompasses a table describing the advantages and disadvantages of various DECT systems

  • Paragraph 4: consider adding twin beam, consider adding dual rotation

Thank you. Both technologies are now outlined in section 4.

  • Paragraph 4: Add references to the described strengths and weaknesses

Thank you. References have been added.

  • Paragraph 5: I seems that DECT and especially the evidence in the literature is highly disappointing. We are 19 years after the introduction of these systems and there is almost no evidence on the added confidence, diagnostic value and patient impact. This probably shows the limitations of the method, that we are provided with a pictorial essay.

Thank you. I agree. The main value for DECT, I believe, is primarily to increase the diagnostic confidence for radiologists to aid clinicians streamline patient management. 

  • Paragraph 5: Several statements have no reference. Please add, if possible, or state “based on our experience”.

Thank you. References are now added. The number of references in the current revised manuscript is 131( compared to 78 previously).

  •  Paragraph 5: It is a fly-through, certainly not a comprehensive overview of clinical problems. What is lacking are the vascular and pulmonary applications, all kinds of benign and malignant diseases in the described regions, iodine contrast boosting and contrast reduction. It is fair to acknowledge this somewhere and lead the readers to other reviews and essays on this topic.

Thank you for your comment. Section 5 now also encompasses pulmonary, cardiac, vascular and oncology sections. The value of low KeV in increasing Iodine attenuation and reducing contrast volume administered is now covered in depth under multiple applications in section 5 and under principles of DECT (Section 3).    

  • Paragraph 5: It feels strange to publish another pictorial essay on dual energy CT, while photon counting CT is already on the market. I would propose to end with a paragraph related to challenges of DECT that maybe solved with photon counting detectors.

Thank you. A "challenges and pitfalls" section delineating limitations of DECT and advantages of Photon-Counting CT over DECT is now incorporated in the manuscript. 

Reviewer 2 Report

Comments and Suggestions for Authors

In this paper, the authors review the utility of dual-source CT in different clinical scenarios. They list various clinical situations and how the DSCT helps to answer the clinical questions.  The review is well-written and easy to read. However, the paper will benefit from having a paragraph about the utility of DSCT in the evaluation of chest pain particularly those with high heart rates that pose a challenge for single source CT 

Author Response

  • In this paper, the authors review the utility of dual-source CT in different clinical scenarios. They list various clinical situations and how the DSCT helps to answer the clinical questions.  The review is well-written and easy to read. However, the paper will benefit from having a paragraph about the utility of DSCT in the evaluation of chest pain particularly those with high heart rates that pose a challenge for single source CT 

Thank you very much for your comments. A subsection for Utilization of DECT in cardiac evaluation including chest pain in now included in the manuscript (Section 5).  

Round 2

Reviewer 1 Report

Comments and Suggestions for Authors

Extensive good revision. Thank you.